

# Missing data imputation using classification and regression trees

Cheng-Yang Chen and Yu-Wei Chang

Department of Statistics, National Chengchi University, Taipei, Taiwan

## ABSTRACT

**Background**. Missing data are common when analyzing real data. One popular solution is to impute missing data so that one complete dataset can be obtained for subsequent data analysis. In the present study, we focus on missing data imputation using classification and regression trees (CART).

**Methods**. We consider a new perspective on missing data in a CART imputation problem and realize the perspective through some resampling algorithms. Several existing missing data imputation methods using CART are compared through simulation studies, and we aim to investigate the methods with better imputation accuracy under various conditions. Some systematic findings are demonstrated and presented. These imputation methods are further applied to two real datasets: Hepatitis data and Credit approval data for illustration.

**Results**. The method that performs the best strongly depends on the correlation between variables. For imputing missing ordinal categorical variables, the *rpart* package with surrogate variables is recommended under correlations larger than 0 with missing completely at random (MCAR) and missing at random (MAR) conditions. Under missing not at random (MNAR), chi-squared test methods and the *rpart* package with surrogate variables are suggested. For imputing missing quantitative variables, the iterative imputation method is most recommended under moderate correlation conditions.

## INTRODUCTION

We encounter missing data when we analyze real data. Simply deleting all elements with missing data results in biased estimates under the missing not at random (MNAR) mechanism (*Rubin, 1976*). Recent studies have either imputed missing data so that the dataset would be complete and common statistical methods could be applied (c.f., *Little & Rubin, 2002*) or proposed models with missing data as one response category (*e.g.*, *Chang, Hsu & Tsai, 2021*; *Debeer, Janssen & De Boeck, 2017*; *Kapelner & Bleich, 2015*; *Twala, Jones & Hand, 2008*). In the present study, we concentrate on the studies that explored imputing missing data.

Suppose there are $N$ elements and $P$ variables in total. Denote $\boldsymbol{X}$ be an $N$ by $P$ data matrix. Assume that the first $P_m$ columns of $\boldsymbol{X}$, denoted as $\boldsymbol{X}_m$, are variables with some missingness, while the remaining $P - P_m$ columns, denoted as $\boldsymbol{X}_c$, are variables with

Corresponding author
Yu-Wei Chang, ychang@nccu.edu.tw

complete data. Imputation can be carried out simply using the mean or mode of a variable or using the mean of all observed data of an element. In contrast, imputation of missing data using assisted parametric or nonparametric models is another popular approach. Classification and regression trees (CART; *Breiman et al., 1984*) are popular modeling approaches developed in recent decades and have also been utilized as assisted models for imputation of missing data. In the present study, we focused on missing data imputation using CART. *Rahman & Islam (2013)* and *Stekhoven & Bühlmann (2012)* suggested a framework in which, for each variable $x_p$, for $p = 1, \ldots, P_m$, we build a binary-splitting CART model with $x_p$ as the response variable and all the other variables as the candidate set of splitting variables (denoted as $\mathcal{S}_p$) for imputing missingness in $x_p$. We follow that framework in the current study. All the elements with observed values for the response variable $x_p$ form the training dataset at the CART building stage. Missing data on $x_p$ are then imputed using the prediction based on the fitted CART model.

When more than one variable has missing data, there might be some missing data on $\mathcal{S}_p$ for the elements in the training dataset. A key difference between different CART-based imputation methods is how they address missing data on $\mathcal{S}_p$. One approach is to exclude elements with some missingness on $\mathcal{S}_p$ at the CART building stage (*Nikfalazar et al., 2020*; *Rahman & Islam, 2013*). A similar idea is to use elements with no missing data on one candidate split variable to compute an impurity measure for the variable (*Kim & Loh, 2001*). Although each candidate split variable may take different sets of elements into account, *Kim & Loh (2001)* found that the approach functioned well on their proposed CART algorithm. As an alternative, several studies have proposed iterative procedures for incorporating elements with missingness on $\mathcal{S}_p$ when building a CART. For each of the variables in $\boldsymbol{X}_m$, *Burgette & Reiter (2010)* suggested building a CART for the variable based on all the "complete variables". Since variables are imputed sequentially in their procedure, imputed variables can be treated as complete when following variables are to be imputed. *Xu, Daniels & Winterstein (2016)* realized the concept of sequential imputation from a different point of view. They decomposed the joint model for the variables in $\boldsymbol{X}_m$ into a sequence of conditional models and built a Bayesian additive regression trees (BART; *Chipman, George & Mcculloch, 2010*) for each conditional model. In contrast, *Stekhoven & Bühlmann (2012)*; *Ramosaj & Pauly (2019)* iteratively imputed variables in $\boldsymbol{X}_m$ based on building random forests (for each variable) until convergence is achieved. They avoided the missingness on $\mathcal{S}_p$ by assigning initial values for all missing data.

An alternative to iterative imputation is to send elements that are missing on a split variable to a child node according to certain criteria. This step is important in imputation since, otherwise, the number of elements in the lower layer decreases so that the CART buildings are less efficient. Several criteria have been proposed under this framework. The majority rule is a common criterion (c.f. *Kim & Loh, 2001*; *Rodgers, Jacobucci & Grimm, 2021*). Elements with missingness on a split variables are sent to a child node with more elements. Another popular method is the surrogate split method (c.f., *Breiman et al., 1984*; *Hapfelmeier, Hothornb & Ulma, 2012*; *Kim & Loh, 2001*; *Quinlan, 1993*; *Therneau & Atkinson, 2009*). In particular, for each split variable, a sequence of surrogate split variables is found based on their similarity to the original split variable. For an element with missing

data on the split variable, those surrogate split variables are considered in order, and the first non-missing surrogate split for the element is used so that the element can be sent to one child node and can be further included in the modeling in the next layer of the CART. Whenever the element is missing on all surrogates, *Therneau & Atkinson (2009)* suggested sending the element to the majority child node. Another interesting point of view is to treat the missingness of one variable as one additional category of the variable (*Kapelner & Bleich, 2015*; *Twala, Jones & Hand, 2008*). When determining some split points, we should further assign the category "missingness" to one of the child nodes.

In the present study, we focus on CART-based methods for imputation of missing data and conducted imputation for variables in turn. We aim to investigate which methods perform better under various conditions. Following *Rahman & Islam (2013)* and *Stekhoven & Bühlmann (2012)*, we considered mixed-type data. In particular, both quantitative and ordinal categorical variables exist in a dataset. We propose a new perspective on missing data imputation using CART and realize the perspective through proposing two resampling-based algorithms for sending elements with missingness on a split variable to one of its child nodes. Algorithm 1 performs random assignments for elements with missing values on the splitting variable to either child node multiple times, and the assignments are further utilized for selecting the best split of its child nodes based on impurity measures, in order to improve CART fittings at the model building stage. Algorithm 2 extends the first algorithm in the following manner. In Algorithm 2, the random assignments at a child node are inherited from the resulting random assignments of its parent node. Following each set of random assignments at the parent node, there is a resulting set of elements at each of child nodes, and the random assignments for the subsequent child nodes depend on the set of elements. The details of the two algorithms are arranged in 'Missing Data Imputation Based on CART'. In addition, we conducted simulation studies to compare existing CART-based imputation methods under various conditions in 'Simulation studies'. For illustration, these methods were further applied to two real datasets: Hepatitis data and Credit approval data, and the results can be found in 'Real data analysis'.

There are interesting findings in our simulation studies. The method that performs the best highly depends on the correlation between variables. The chi-squared tests in *Loh & Shih (1997)* and *Kim & Loh (2001)* in combination with the two proposed algorithms achieve better imputation accuracy when correlations between quantitative variables are around 0. The multiple imputation by chained equations (MICE; *Raghunathan et al., 2001*) method and the R package *rpart* with surrogate variables generally work better under correlations $\rho > 0$ with missing completely at random (MCAR) and missing at random (MAR) conditions, when we impute missing ordinal categorical variables. Under MNAR, methods based on the chi-squared tests methods are recommended. For imputing missing quantitative variables, the iterative imputation method in *Stekhoven & Bühlmann (2012)* in a CART version and the MICE method generally perform better under moderate correlation conditions. These findings provide some practical guidelines for choosing a good imputation method with higher imputation accuracy in future real data analysis.

**Algorithm 1** a resampling-based (RE) algorithm for the elements with missing values on a subsequent split variable

**Input:** data matrix $X$; a variable to be imputed: $x_p$; some missingness on variable $x_q$.

**Initialization:** Start with the root node $t$ where it has been determined to be further split, and its split variable has been determined as $x_q$. Suppose that $V_{t,q} \neq \varnothing$. If some child node of $t$, $t_L$ or $t_R$, is to be further split, consider the followings.

**Step 1:** (repeat resamplings $H$ times)

    **for** $h$ in $1:H$ **do**

        **for** $b$ in $V_{t,q}$ **do** ($b$ is an element at node $t$)

            Randomly assign $b$ to $t_L$ or $t_R$;

        **end for**

    $V_{t,q,h,L} \leftarrow$ all elements in $V_{t,q}$ that are assigned to $t_L$;

    $V_{t,q,h,R} \leftarrow V_{t,q} \setminus V_{t,q,h,L}$;

    Refer the results of the sampling as $B_h$.

    **end for**

**Step 2:** (Determine the split variables and split points for $t_L$)

    **for** $h$ in $1:H$ **do**

        Compute an impurity measure based on both (i) $W_{t,q,L}$, defined as the set of the elements assigned to $t_L$ according to $x_q$ and (ii) $V_{t,q,h,L}$.

    **end for**

    Choose the variable, denoted as $x_{q_L}$, with the smallest impurity measure among all $\mathcal{S}_p$ and $B_h$, $h = 1, \dots, H$, as the split variable for node $t_L$.

    Split points are determined similarly.

    Split variables and split points for $t_R$ are determined similarly.

**Step 3:** (the issues of sending elements to child nodes for node $t_L$)

    **for** $b$ in $W_{t,q,L}$ **do**

        **if** $b$ is missing on $x_{q_L}$ **do**

            Randomly assign $b$ to $t_{L_L}$ or $t_{L_R}$, the child nodes of $t_L$.

        **end if**

    **end for**

    Deal with the issues of sending elements to child nodes for $t_R$ similarly.

**Step 4:** Consider $t \leftarrow t_L$ and $t \leftarrow t_R$, respectively, and then repeat Steps 2 and 3 until no end nodes is to be further split.

**Output:** a CART built for $x_p$, and all elements in the training dataset of $X$ being sent to one of end nodes.

## MATERIALS AND METHODS

### Missing data imputation based on CART

In the current section, we first introduce two proposed resampling algorithms in 'Resampling-based CART imputation methods'. We will then outline the CART-based missing data imputation methods in the literature that we compare in 'Other CART-based imputation methods'.

**Algorithm 2** a hierarchical resampling-based (H-RE) algorithm for the elements with missing values on a subsequent split variable

**Input:** data matrix $X$; a variable to be imputed: $x_p$; some missingness on variable $x_q$.

**Initialization:** Same as Algorithm 1.
**Step 1:** Same as Algorithm 1.
**Step 2:** Same as Algorithm 1.
**Step 3:** (the issues of sending elements to child nodes for node $t_L$)

The resampling at node $t_L$ is based on the results of the resampling at its parent node $t$.

**for** $h$ in $1:H$ **do**

Based on the $h$-th resampling of node $t$ in Step 1, the set of elements at node $t_L$ is $V_{t,q,h,L} \cup W_{t,q,L}$.

Determine child nodes for elements in $V_{t,q,h,L} \cup W_{t,q,L}$ with respect to the split variable $x_{q_L}$. For the elements which are in $V_{t,q,h,L} \cup W_{t,q,L}$ and are missing on $x_{q_L}$, conduct resampling once:

> **for** $b$ in $V_{t,q,h,L} \cup W_{t,q,L}$ **do**
>> **if** $b$ is missing on $x_{q_L}$ **do**
>>> Randomly assigned $b$ to $t_{L_L}$ or $t_{L_R}$, the left or right child node of $t_L$;
>> **end if**
> **end for**
> $V_{t_L,q_L,h,L_L} \leftarrow$ all elements assigned to $t_{L_L}$ at this stage;
> $V_{t_L,q_L,h,L_R} \leftarrow V_{t,q,h,L} \cup W_{t,q,L} \setminus V_{t_L,q_L,h,L_L}$.

**end for**

Deal with the issues of sending elements to child nodes for $t_R$ similarly.
**Step 4:** Consider $t \leftarrow t_L$ and $t \leftarrow t_R$, respectively, and then repeat step 2 and 3 until no end nodes is to be further split.
**Output:** a CART built for $x_p$, and all elements in the training dataset of $X$ being sent to one of end nodes.

For a data matrix $X$, variables with missing data are imputed in turn. For each variable $x_p$ in $X_m$, we build a classification tree when $x_p$ is an ordinal categorical variable. When $x_p$ is a quantitative variable, a regression tree is built. The steps for building a CART include determining a split variable, selecting a split point, assigning elements to child nodes, and determining stopping rules. The classical CART fitting procedure involves determining split variables and split points *via* the Gini index (GI) for categorical response variables $x_p$ and the residual sum of squares (RSS) for continuous response variables $x_p$ (c.f., *Breiman et al., 1984*; *James et al., 2013*). As noted in *Loh & Shih (1997)* and *Kim & Loh (2001)*, several earlier algorithms for determining split variables resulted in selection bias, in the sense that some algorithms preferred categorical split variables, others preferred continuous split variables, and algorithms tended to select split variables with more missing data. To solve the selection bias problem, *Loh & Shih (1997)* and *Kim & Loh (2001)* suggested to separate the determination of a split variable and some split points into two steps and proposed the

use of a chi-squared test to determine a split variable (referred to as the CHI method in the present study). In particular, for each variable in $\mathcal{S}_p$, a chi-squared test is conducted with respect to $x_p$. A smaller $p$-value of the test indicates some higher relationship between the variable and $x_p$; thus, the variable with the smallest $p$-value is the suggested split variable. A numerical variable should first be discretized into four categories, and the chi-squared test can be applied. For a given split variable based on the chi-squared test, the split points are determined through the usual GI or RSS methods.

### Resampling-based CART imputation methods

Consider a node $t$ where it has been determined to be further split, and its split variable has been determined as $x_q$. Denote $V_{t,q}$ as the collection of all elements at node $t$ with missingness on $x_q$. When the number of elements in $V_{t,q}$ is large, the CART constructions at lower layers are not efficient since few elements are available at the nodes of lower layers. It is important to address such missing data in CART constructions. In the current section, we introduce our new perspective on missing data in a CART imputation problem and explore how to realize this perspective through two resampling algorithms.

When an element is not missing on the splitting variable $x_q$, the observed data lead the element to either the left child node of $t$ (denoted as $t_L$) or the right child node of node $t$ (denoted as $t_R$). In other words, one of them is the truth. On the other hand, when an element is missing on the splitting variable, the true underlying value, if the value could be observed, still leads the element to either child node. This idea motivated us to conduct the random assignment in the current study. For each of the elements in $V_{t,q}$, we randomly assign it to the $t_L$ or $t_R$ child node, and some of the assignments would be *correct*. The advantage of sending elements with missingness on split variables to child nodes is that as many elements as possible are retained for CART buildings in the lower layers of a tree. We implement the idea in Algorithms 1 and 2.

Algorithm 1 (RE Algorithm):

In Algorithm 1, we repeat random assignments for elements in $V_{t,q}$ $H$ times so that those elements with missing splitting variables are still utilized in determining some split variable and split point for child node $t_L$ (or $t_R$). Hopefully, those *correct* assignments among $H$ random assignments could lead to better fitted CART models. That is, the split variable for the child node of $t$ can be determined with a better impurity measure. In the numerical experiments in 'Simulation studies' and 'Real data analysis', we set $H = 20$.

Since all the elements in $V_{t,q}$ are sent to one of the child nodes, $t_L$ or $t_R$, they can be incorporated at the stage of determining split variables for $t_L$ or $t_R$. The determination of split variables is still based on common impurity measures. In determining a split variable for $t_L$ or $t_R$, since we have $H$ random assignments, the impurity measures are evaluated over all $H$ random assignments and over all the candidate sets $\mathcal{S}_p$. The methods for determining the split points for $t_L$ or $t_R$ are the same as usual CART algorithms. The details are arranged in the table for Algorithm 1. In Step 2, to determine the split variables for $t_L$, compute $H$ impurity measures for one candidate split variable. In particular, the $h$-th measure, for $h = 1, \ldots, H$, is based on both (i) $W_{t,q,L}$, defined as the set of the elements assigned to $t_L$ according to $x_q$ and (ii) $V_{t,q,h,L}$, those sent to $t_L$ according to resampling. There are some

remarks for Step 3: when elements in $W_{t,q,L}$ is missing on $x_{q_L}$ (i.e., $V_{t_L,q_L}$), we cannot send these elements to child nodes of $t_L$, and therefore, we have to repeat the resampling procedure in Step 1 with respect to $V_{t_L,q_L} \cup V_{t,q}$. Besides, the resampling set for a node $t_L$ should be the union of the sets $V's$ of all the parent nodes of $t_L$. In the following, we refer to this resampling-based algorithm as the RE algorithm, where RE is the abbreviation for resampling. The idea of randomly sending elements which are missing on one split variable to child nodes is similar to the on-the-fly imputation (OTFI) method in *Ishwaran et al. (2008)*. The differences between the OTFI and RE algorithms are as follows: we repeated the random assignment $H$ times and search for the best assignment at the stage of determining the splitting of its child nodes, while the OTFI method conducted the random assignment once. In addition, we implemented the idea of random assignment in CART settings while *Ishwaran et al. (2008)* and *Tang & Ishwaran (2017)* conducted some random assignment in a random forest model.

Algorithm 2 (H-RE Algorithm):

In the RE algorithm, the resampling for node $t$ and the resampling for the child node $t_L$ (or $t_R$) are conducted separately. That is, the resulting resamplings of a node $t$ are not inherited by its child nodes. We further consider certain hierarchy between the resamplings for node $t$ and that for its child nodes in Algorithm 2. Algorithm 2 is referred to as a hierarchical-resampling algorithm, abbreviated as the H-RE algorithm.

Steps 1 and 2 of Algorithm 2 are the same as those in Algorithm 1. The resampling at one node is nested in one previous layer of the tree. For the resampling at node $t_L$, we simply trace the $H$ resampling results $V_{t,q,h,L}$, $h = 1, \ldots, H$, of its parent node, $t$. For $V_{t,q,h,L}$, we take the union with $W_{t,q,L}$, the set of elements assigned to $t_L$ according to $x_q$. We concentrate on the elements in $V_{t,q,h,L} \cup W_{t,q,L}$ with missing values on $x_{q_L}$, and resampling is conducted once for each $h$. The results are assigned to child nodes $t_{L_L}$ and $t_{L_R}$. There are $H$ resamplings at Step 3, and the candidate sets to be resampled, $V_{t,q,h,L} \cup W_{t,q,L}$ with missing on $x_{q_L}$, in the $H$ resamplings differ between resamplings. The elements in $V_{t_L,q_L,h,L_L}$ are further utilized in determining the split variables for $t_{L_L}$ and $t_{L_R}$. The procedure is summarized in Algorithm 2, referred to as the H-RE algorithm.

### *Other CART-based imputation methods*

One purpose of the study is to compare CART-based imputation methods and explore which methods perform better. In this section, we outline all the methods implemented in the current study. Some of them are newly proposed methods in 'Resampling-based CART imputation methods', and some are existing methods in the literature or package. For ease of mention, they are referred to as $M_1$ to $M_9$ in the following.

The R package *rpart*, developed based on *Breiman et al. (1984)*, is a popular package for building CART models. In the package, an element missing on its subsequent split variable is sent to one of the child nodes based on the popular surrogate variable method (*Breiman et al., 1984*; *Hapfelmeier, Hothornb & Ulma, 2012*; *Kim & Loh, 2001*, *Quinlan, 1993*; *Therneau & Atkinson, 2009*).

$M_1$: R package *rpart* with surrogate variables.

As mentioned in the beginning of 'Missing data imputation based on CART', the GI (RSS) and CHI are two common methods for selecting split variables and split points. According to the literature and what we proposed in 'Resampling-based CART imputation methods', we have three methods for sending elements which are missing on a split variable to one of the child nodes: (i) majority rule (c.f. *Rodgers, Jacobucci & Grimm, 2021*); (ii) the RE algorithm; and (iii) the H-RE algorithm. In combination, we have the six following CART-based methods for imputing missing data:

$M_2$: GI (RSS) + majority rule,

$M_3$: GI (RSS) + RE algorithm,

$M_4$: GI (RSS) + H-RE algorithm,

$M_5$: CHI + majority rule,

$M_6$: CHI + RE algorithm,

$M_7$: CHI + H-RE algorithm.

For a fair comparison with previous CART-based imputation methods, we implemented the iterative imputation method in *Stekhoven & Bühlmann (2012)* in a CART version; this method is referred to as $M_8$. In particular, in $M_8$, each variable in $X_m$ is iteratively imputed based on the construction of CART models, where all the other variables in $X$ are potential splitting variables. The procedures iterate until the imputations are the same between iterations for each missingness. With proper initials, we do not have to worry about the missingness on $S_p$ when we build a CART for imputing $x_p$ since all variables are imputed in the previous iteration; therefore, all imputed values together with observed data form "complete data" for $S_p$.

$M_8$: Implementing the iterative imputation method in *Stekhoven & Bühlmann (2012)* in a CART version.

In addition to the above eight imputation methods, MICE is another popular method, and it was also incorporated in our comparison[1]. As an extension of the multiple imputation method (*Rubin, 1987*), MICE (*Raghunathan et al., 2001*) was capable to deal with data sets where more than one variable with missingness. *Burgette & Reiter (2010)* suggested to adopt CART as the assisted model in the MICE algorithm, and we included this setting for comparison since its popularity. *Wang et al. (2022)* found that MICE with CART outperformed MICE with random forest and MI with some other machine learning methods in terms of imputation accuracy in survey data and some simulated data. *Akande, Li & Reiter (2017)* focused on categorical data and found that MICE with CART obtained better imputation accuracy than MICE with generalized linear models. This method is called $M_9$, and it is implemented using the R package *mice* with the *cart* option.

$M_9$: R package *mice* with the *cart* option.

Simulation studies were conducted to compare these methods. The simulation designs are illustrated in 'Simulation Studies', and results are demonstrated in 'Results for simulation studies'.

## Simulation studies

Two questions addressed by these simulation studies are as follows: (i) Which imputation method results in better imputation accuracy than other methods? (ii) Does the

[1] As suggested by one reviewer.

performance depend on the conditions of the data? To answer the second question, we examined the effects of the correlation between variables and missing mechanisms on the efficiency of missing data imputation. The simulation settings are outlined in 'Simulation settings', and the results of the simulations are presented in 'Results for simulation studies'. The advantage of conducting the simulation in 'Simulation studies' is that we simulated the observed data ourselves, and thus could control the correlations between variables. In this way, we were able to demonstrate the effects of correlations on the efficiency of imputation.

### Simulation settings

The number of variables was fixed at 15 (*Ramosaj & Pauly, 2019*), and the number of observations was fixed at 500. Following *Beaulac & Rosenthal (2020)*; *Rahman & Islam (2013)*, and *Stekhoven & Bühlmann (2012)*, we studied the case in which both quantitative and ordinal categorical variables are presented in a dataset. Variables 1 to 8 are quantitative, and variables 9 to 15 are ordinal categorical. Since there were many aspects to be examined, we considered three simulation studies, and each of studies had its purpose. We illustrate the three studies in the followings.

*Study 1:* In Study 1, we aimed to investigate whether some crucial aspects have impacts on imputation accuracy, and which imputation methods performed better under various conditions. The following factors were considered. (i) There are 4 different correlations between variables. (ii) There are three types of missing mechanisms: MCAR, MAR, and MNAR. (iii) There are missing data on two types of variables: quantitative and ordinal categorical variables (*Ramosaj & Pauly, 2019*).

There are a total of $4 \times 3 \times 2$ conditions.

To generate the data, there were four settings for the correlations between variables: 0, $\pm 0.3$, $\pm 0.5$, and $\pm 0.7$. Take the setting $\pm 0.5$ as an example; the correlation between each pair of variables was approximately $\pm 0.5$. For brevity, we used $\rho = 0.5$ for the case $\rho = \pm 0.5$ hereafter. Following *Ramosaj & Pauly (2019)*, the quantitative variables were generated through a multivariate normal distribution. The ordinal categorical variables were generated *via* the standard approach, in which there were some underlying continuous variables and the observed ordinal categorical variables were some discretization of the continuous variables. The settings of correlations regarding ordinal categorical variables indicate (i) the correlations between underlying continuous variables and (ii) the correlations between one underlying continuous variable and one variable among variables 1 to 8. Among the seven ordinal categorical variables, there were five binary variables and two variables with three categories. The ratios between categories for the seven variables were 6:4, 6:4, 6:4, 5:5, 5:5, 3:3:4, and 4:3:3.

To generate missingness in the above simulated data, we considered MCAR, MAR, and MNAR (*Rubin, 1976*). For the missingness of one variable $Y_1$, let $y_{i1}$ be the observation of subject $i$ on variable $Y_1$, and let $Z_i$ be a binary indicator denoting whether $Y_{i1}$ is missing ($Z_i = 1$) or not ($Z_i = 0$). Denote $Y_2$ and $Y_3$ as other variables in the dataset. When

$$P\left(Z_i = 1 | y_{i1}, y_{i2}, y_{i3}\right)$$

does not depend on the value of $y_{i2}$ and $y_{i3}$, $Y_1$ is called MCAR. In contrast, MAR on $Y_1$ refers to the case that its missingness depends on the value of some other variables, say $y_{i2}$ and $y_{i3}$, but does not depend on its own value $y_{i1}$. That is,

$$P\left(Z_i = 1 | y_{i1}, y_{i2}, y_{i3}\right) = P\left(Z_i = 1 | y_{i2}, y_{i3}\right) \text{ for all subject } i.$$

MNAR on $Y_1$ indicates the case that its missingness depends on its own value, $y_{i1}$, but does not depend on the value of some other variables, say $y_{i2}$ and $y_{i3}$. Specifically,

$$P\left(Z_i = 1 | y_{i1}, y_{i2}, y_{i3}\right) = P\left(Z_i = 1 | y_{i1}\right) \text{ for all subject } i.$$

We considered missing data on quantitative and ordinal categorical variables. Missing quantitative variables were set at variables 1, 3, 5, 7, and 8. Missing ordinal categorical variables were set at variables 9, 11, 12, 13, and 15. The missing rate was controlled at 10% in Study 1. That is, for each of the above variables, there were 10% missing data.

To generate MCAR, each observation was determined missing-or-not, independently with other observations, with missing rate 10%. To generate missing data under the MAR mechanism, the missingness of the above 10 variables was set to depend on variable 2, 4, 2, 2, 4, 6, 6, 4, 6, and 2, respectively; the smaller the values on variable 2 (or 4, 6), the higher the probabilities of missingness on the above 10 variables. To generate MNAR mechanism on one categorical variable, the probability of missing was set to be higher on its first category. For MNAR mechanism on one quantitative variable, the missing probability was set to be higher on its first 30% percentiles.

For each of the $4 \times 3 \times 2$ conditions, we generated 100 replicates (*i.e.*, 100 datasets) independently. Each of the datasets was imputed respectively using the 9 methods in 'Other CART-based imputation methods'.

*Study 2:* In Study 1, the missing rate was controlled at 10% in various conditions. We further manipulated the missing rate as a factor in Study 2, and we investigated whether the phenomenon observed in Study 1 remained the same under different missing rates. Three levels of missing rates, 10%, 20%, and 30%, were considered in Study 2, and we focused on quantitative variables.

*Study 3:* In Study 3, we investigated whether the rankings of accuracy for different methods showed similar patterns under different sample sizes. In Study 3, we further examined sample sizes equivalent to 500 and 1,000, and we focused on the following conditions: missing rate 10% and missing on quantitative variables.

All the simulations were conducted using the *R* software (*Beaulac & Rosenthal, 2020*), and the simulation results are presented in the next section.

## RESULTS

### Results for simulation studies

The imputation performances for quantitative and ordinal categorical variables were summarized so that we were able to compare the imputation accuracy across methods. For quantitative variables, we evaluated the imputation performance by comparing the imputed values to *true data* (the values in the generated data) in terms of the Root Mean Squared Error (RMSE). For $x_{i,p}$, the value of Subject $i$ on variable $p$, let $\hat{x}_{i,p}$ denote its

imputation. The RMSE for variable $X_p$ for one dataset is

$$RMSE_p = \sqrt{\frac{1}{|D_p|}\sum_{i\in D_p}\left(x_{i,p} - \hat{x}_{i,p}\right)^2},$$

where $D_p$ is the collection of indices of subjects who are missing on variable $X_p$ in the whole dataset and $|D_p|$ is the total number of elements in collection $D_p$. The smaller $RMSE_p$ is, the better the method.

For ordinal categorical variables, we summarized the imputation accuracy *via* the proportion of correct classifications (PCC). The PCC for variable $X_p$ for one dataset is

$$PCC_p = \frac{1}{|D_p|}\sum_{i\in D_p}I_{\{x_{i,p}=\hat{x}_{i,p}\}},$$

where $I_{\{\cdot\}}$ is the indicator function. The larger $PCC_p$ is, the better the method.

We summarized the imputation performance for quantitative and ordinal categorical variables in the following two subsections. An interesting founding was that the imputation accuracy not only differed among methods $M_1$ to $M_9$ but also differed to a certain degree between variables. Therefore, we summarized the imputation performance for each of the 10 variables separately. That is, for each of the 10 variables, $RMSE_p$ or $PCC_p$ was averaged over the 100 replicates.

### Imputation performance for quantitative variables

The imputation performances for variables 1, 3, 5, 7, and 8, the quantitative variables with missing values, are organized in Figs. 1 to 4. In particular, we arranged $RMSE_p$ for the same $\rho$ across all MCAR, MAR, and MNAR conditions in one figure. For example, all the results under $\rho =0$ are given in Fig. 1. To obtain better visualization for distinguishing the rankings between methods, all $RMSE_p$ values under one method are linked using a line. In particular, under each variable and each method, we averaged $RMSE_p$ over 100 replicates. The mean and standard deviation, taken over 100 replicates, of $RMSE_p$ for each variable and each method are reported in the Tables S1 to S4. In the followings, we report the simulation results and outline the method(s) with the best or better imputation accuracy.

We include MICE in the comparison since it is a popular method. As remarked by *Hapfelmeier, Hothornb & Ulma (2012)*, the imputation of MICE is based on multiple trees and takes the average of them while the imputations by other methods are based on single tree. On the basis of the argument, they noted that it was not quite fair to compare MICE to single-tree methods after their comparison. *Stekhoven & Bühlmann (2012)* made another important remark regarding MICE: though MICE was popular, its strength was to provide some uncertainty measure of imputation, and it was not primarily designed for constructing single imputation value. Based on these reasons, we arrange the presentation of 'Imputation performance for quantitative variables' and 'Imputation performance for ordinal categorical variables' in the following manners. We first discuss results about MICE ($M_9$) *versus* all the other eight methods ($M_1$ to $M_8$) to present an overall view on how the single-tree methods perform, compared with the multiple-trees method (MICE). We then narrow the comparison to single-tree methods $M_1$ to $M_8$ for a fair comparison.

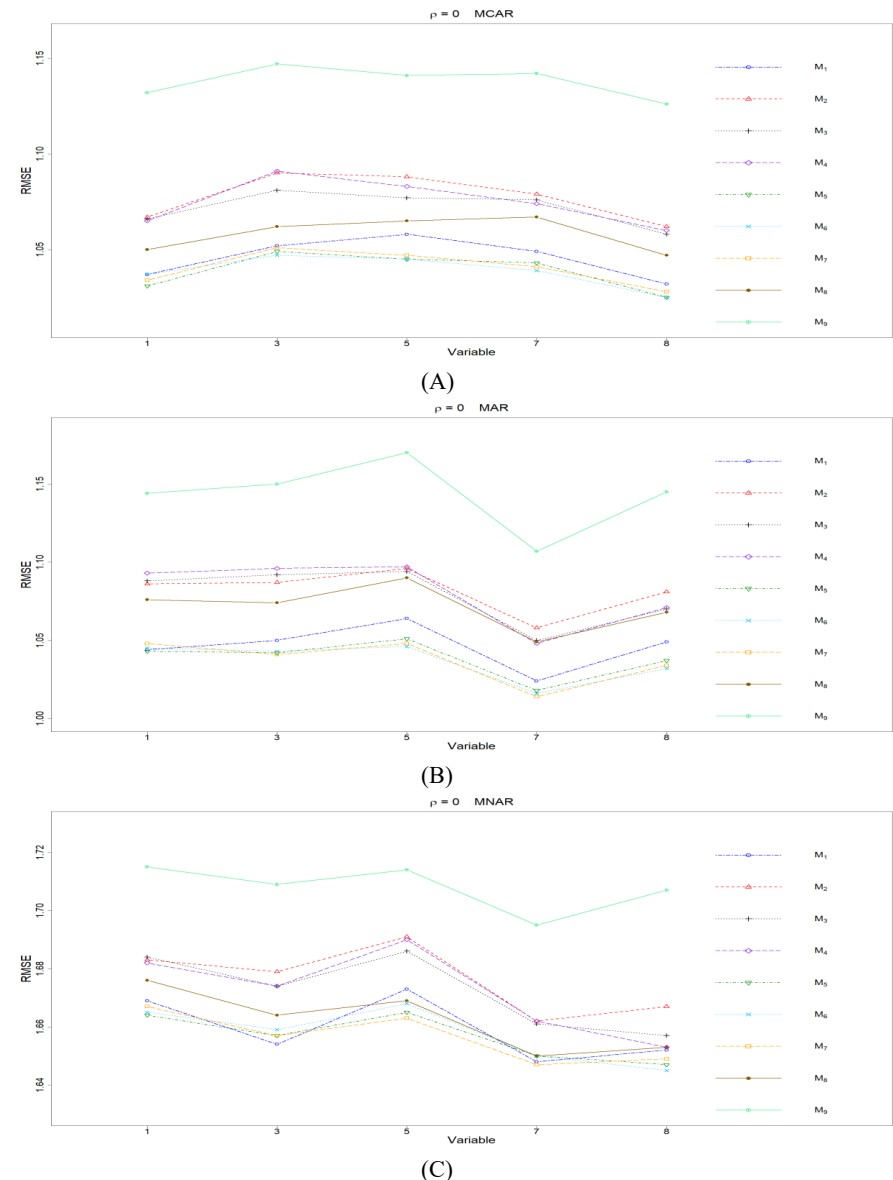

**Figure 1** **The imputation performances for quantitative variables 1, 3, 5, 7, and 8 under the condition correlation = 0, missing rate 10%, $N = 500$, and (A) MCAR, (B) MAR, (C) MNAR.** The $RMSE_p$ under methods $M_1$ to $M_9$ are plotted respectively for each of the variables. Imputation methods: $M_1$: *rpart* with surrogate variables; $M_2$: GI (RSS) + majority rule; $M_3$: GI (RSS) + RE algorithm; $M_4$: GI (RSS) + H-RE algorithm; $M_5$: CHI + majority rule; $M_6$: CHI + RE algorithm; $M_7$: CHI + H-RE algorithm; $M_8$: iterative imputation; $M_9$: *mice* with cart option.

Overall speaking, for all nine methods involving the three types of missing mechanisms, the imputation accuracy increases as the correlation $\rho$ increases; this phenomenon is expected since the greater the correlation between variables is, the more information can be borrowed from each other when searching for efficient split variables for imputation; thus, imputation accuracy should increase with correlation.

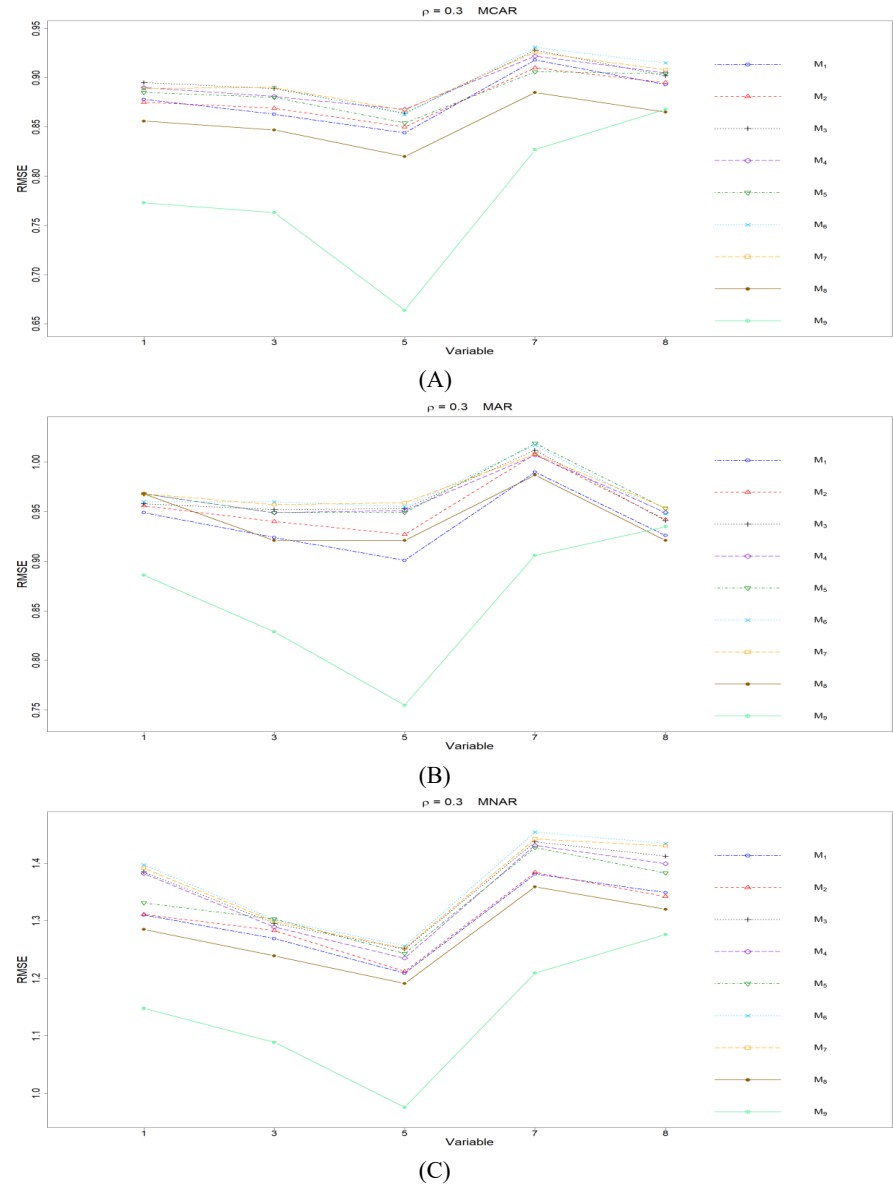

**Figure 2** **The imputation performances for quantitative variables 1, 3, 5, 7, and 8 under the condition correlation = 0.3, missing rate 10%, $N = 500$, and (A) MCAR, (B) MAR, (C) MNAR.** The $RMSE_p$ under methods $M_1$ to $M_9$ are plotted respectively for each of the variables. Imputation methods: $M_1$: *rpart* with surrogate variables; $M_2$: GI (RSS) + majority rule; $M_3$: GI (RSS) + RE algorithm; $M_4$: GI (RSS) + H-RE algorithm; $M_5$: CHI + majority rule; $M_6$: CHI + RE algorithm; $M_7$: CHI + H-RE algorithm; $M_8$: iterative imputation; $M_9$: *mice* with cart option.

First, we present the comparison between MICE ($M_9$) and all the other eight methods ($M_1$ to $M_8$). According to Fig. 1, $M_9$ generally works worse than $M_1$ to $M_8$ when there is no correlation ($\rho = 0$), and the phenomena remain the same across the three types of missing mechanisms. $M_9$ has some advantage when the correlation between variables is

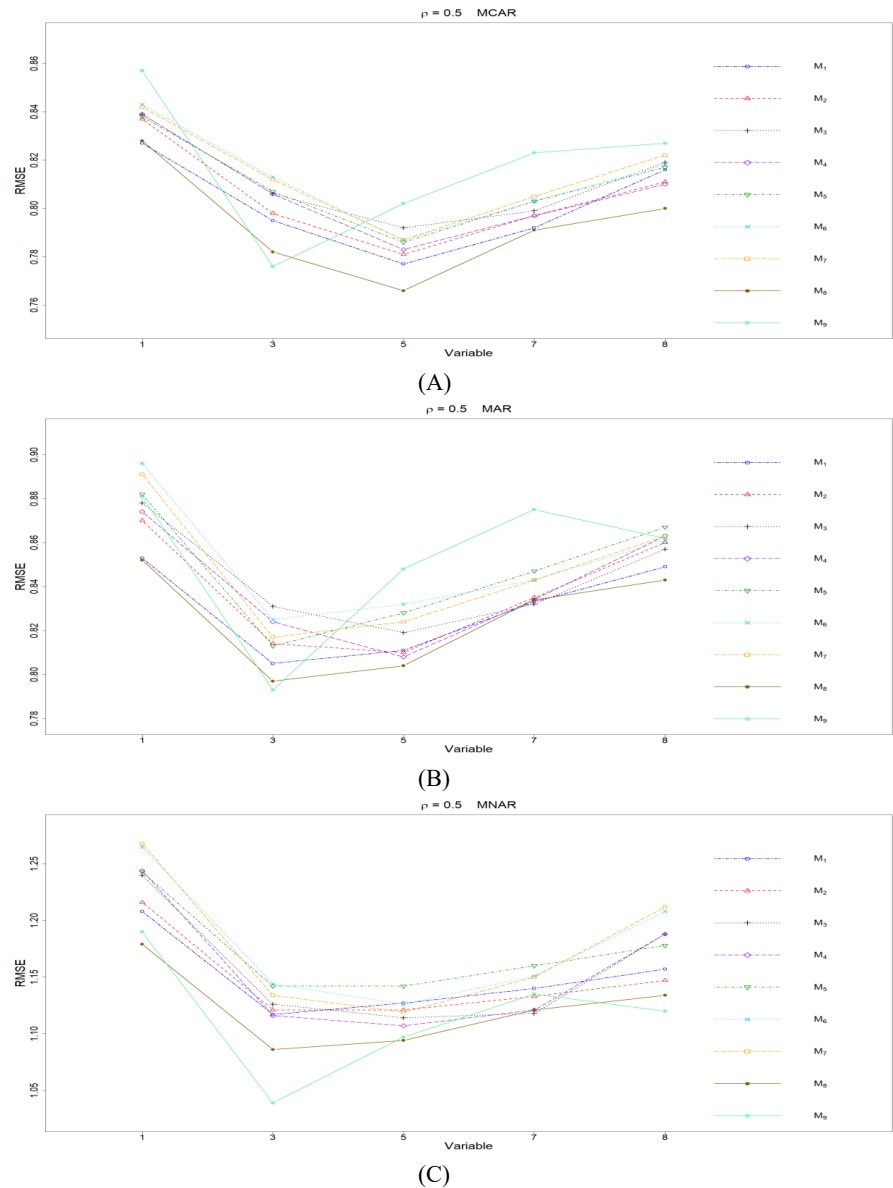

**Figure 3** **The imputation performances for quantitative variables 1, 3, 5, 7, and 8 under the condition correlation = 0.5, missing rate 10%, N = 500, and (A) MCAR, (B) MAR, (C) MNAR.** The $RMSE_p$ under methods $M_1$ to $M_9$ are plotted respectively for each of the variables. Imputation methods: $M_1$: *rpart* with surrogate variables; $M_2$: GI (RSS) + majority rule; $M_3$: GI (RSS) + RE algorithm; $M_4$: GI (RSS) + H-RE algorithm; $M_5$: CHI + majority rule; $M_6$: CHI + RE algorithm; $M_7$: CHI + H-RE algorithm; $M_8$: iterative imputation; $M_9$: *mice* with cart option.

low ($\rho = 0.3$). For the conditions with higher correlations ($\rho = 0.5$ or $0.7$), there is no clear pattern whether $M_9$ works better than $M_1$ to $M_8$ or not.

Second, we concentrate on all the single-tree methods ($M_1$ to $M_8$) for a fair comparison. An interesting and important finding is that which method performs the best depends on

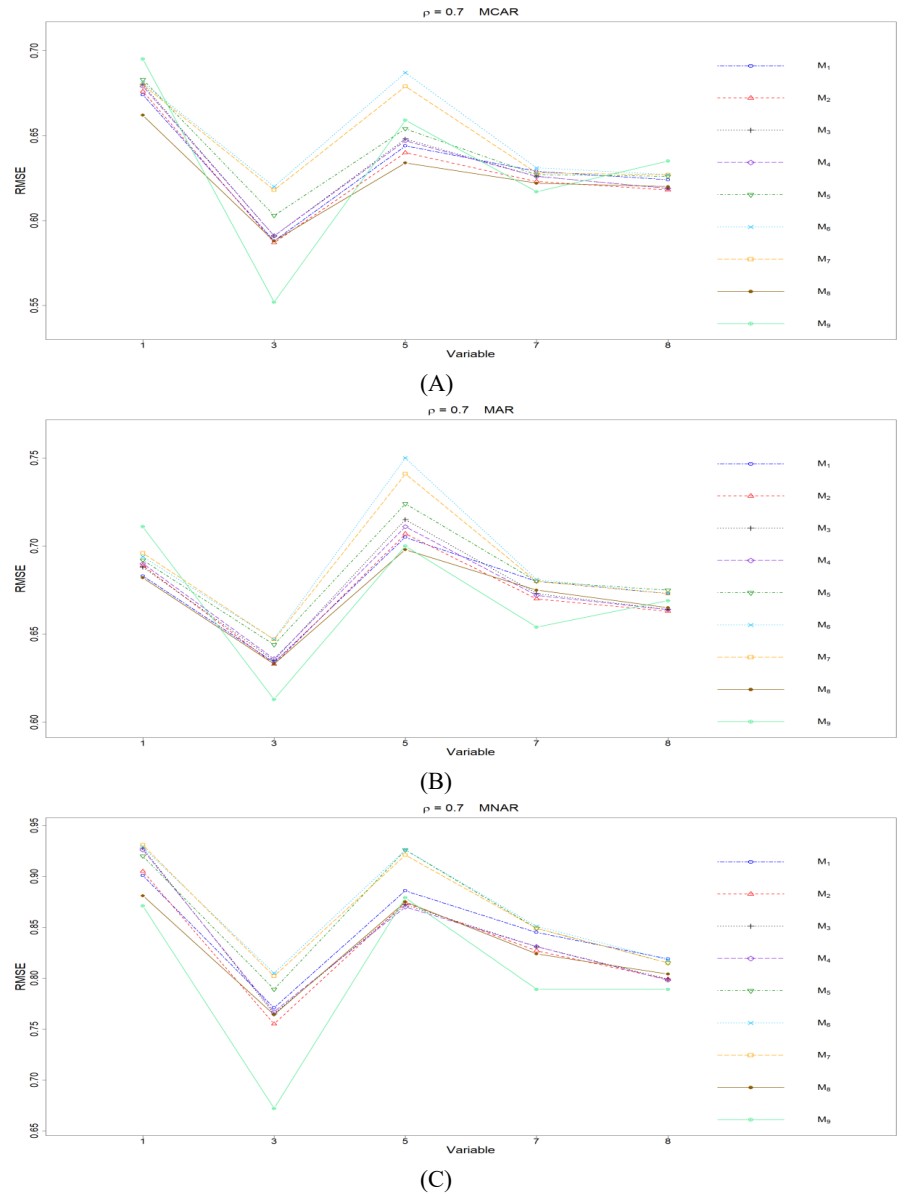

**Figure 4  The imputation performances for quantitative variables 1, 3, 5, 7, and 8 under the condition correlation = 0.7, missing rate 10%, $N = 500$, and (A) MCAR, (B) MAR, (C) MNAR.** The $RMSE_p$ under methods $M_1$ to $M_9$ are plotted respectively for each of the variables. Imputation methods: $M_1$: *rpart* with surrogate variables; $M_2$: GI (RSS) + majority rule; $M_3$: GI (RSS) + RE algorithm; $M_4$: GI (RSS) + H-RE algorithm; $M_5$: CHI + majority rule; $M_6$: CHI + RE algorithm; $M_7$: CHI + H-RE algorithm; $M_8$: iterative imputation; $M_9$: *mice* with cart option.

the correlation between variables. When there is no correlation ($\rho = 0$), the CHI methods ($M_5$ to $M_7$) generally perform better than the other five methods, according to Fig. 1. *Loh & Shih (1997)* and *Kim & Loh (2001)* remarked that their proposed CHI method is selection unbiased when there is no correlation among variables and when there are no missing

data in the dataset. Here, we further found out that the CHI methods ($M_5$ to $M_7$) also outperform the other five methods in terms of imputation accuracy when missing data are present; this advantage is related to the nice properties of selection unbiasedness, where all the other variables are equally selected as split variables when there is no correlation between them. Better selection of split variables by the CHI methods at the training stage could result in better predictions at the testing stage, and the predictions are adopted as imputed values in the CART imputation framework. This is our interpretation about the results of CHI methods. Among $M_5$ to $M_7$ under $\rho = 0$, the RE or the H-RE algorithm ($M_6$ and $M_7$) performs slightly better than the majority rule ($M_5$) for sending elements to a child node. We suspect that the advantage of the CHI methods with the RE and the H-RE algorithms is related to that the two algorithms search across many possible assignments and choose the one with best impurity measure so that some better predictions are attained under the condition of $\rho = 0$.

When some correlation exists between variables and under $\rho = 0.3$ and $0.5$, the iterative imputation method ($M_8$) generally outperforms the other seven single-tree methods ($M_1$ to $M_7$) across all the five variables and the three types of missing mechanisms, based on Figs. 2 and 3. There are some nice interpretations about the results. The iterative imputation method not only tries to extract information (the middle correlation between variables) from other variables when building a CART but also improves some imputation through iterations. According to Figs. 2 to 4, the iterative improvement in $M_8$ is important when the correlation between variables is in the moderate ($\rho = 0.3$ and $0.5$). In such cases, $M_2$ to $M_4$ (the GI/RSS methods, majority rules, and the resampling strategies) might not impute missing data as effectively as iterative imputation method ($M_8$). In contrast, the iterative improvement does not improve much when the correlation is as high as $\rho = 0.7$ since the high correlation between variables can be fully utilized in building one CART and iterations are not necessary. In addition, when $\rho = 0.7$, the *rpart* package with surrogate variables ($M_1$), $M_2$ to $M_4$, and $M_8$ function equally well, and none of them has an apparent advantage over the others. What is more, the common method, $M_1$, also yields fair performance under moderate correlations.

### *Imputation performance for ordinal categorical variables*

The imputation performances for ordinal categorical variables are presented in this subsection. Some missing data exist on variables 9, 11, 12, 13, and 15, and the imputation accuracy is summarized in Figs. 5 to 8 for conditions $\rho = 0, 0.3, 0.5$, and $0.7$. Since there are simply 2 or 3 categories, some methods achieve the same imputation results in many replicates; therefore, the $PCC_p$s are similar across many methods. The imputation accuracy for missing ordinal categorical variables under the nine methods is rather close to each other, compared to the imputation accuracy for missing quantitative variables. In contrast, the imputation accuracy differs greatly between variables, and the variables differ in the number of categories and the ratios between categories.

First, we compare the imputation performance between MICE ($M_9$) and all the other eight methods ($M_1$ to $M_8$). According to Figs. 5 to 8, the imputation accuracy of $M_9$ is generally worse, or sometimes about the same, than $M_1$ to $M_8$, across all the correlations

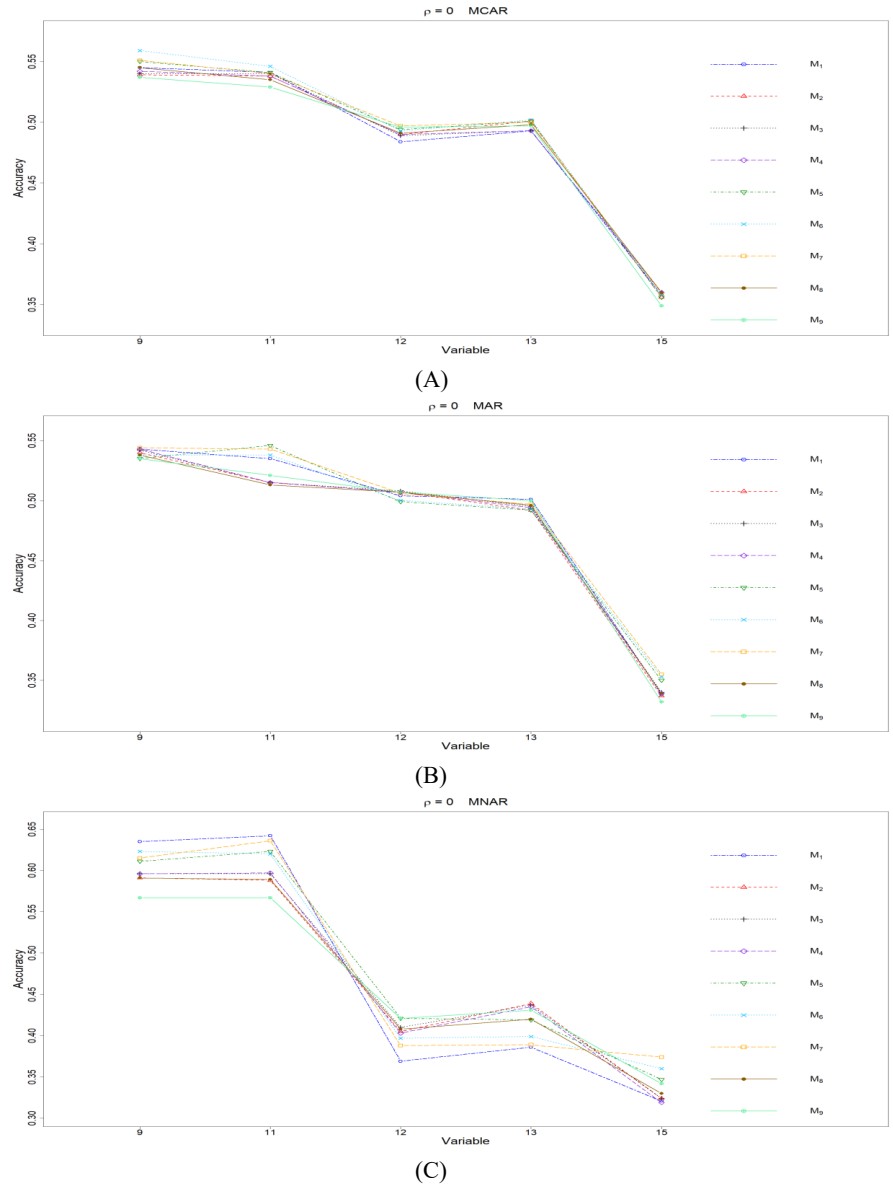

**Figure 5** **The imputation performances for ordinal categorical variables 9, 11, 12, 13, and 15 under the condition correlation = 0, missing rate 10%, $N = 500$, and (A) MCAR, (B) MAR, (C) MNAR.** The $PCC_p$ under methods $M_1$ to $M_9$ are plotted respectively for each of the variables. Imputation methods: $M_1$: *rpart* with surrogate variables; $M_2$: GI (RSS) + majority rule; $M_3$: GI (RSS) + RE algorithm; $M_4$: GI (RSS) + H-RE algorithm; $M_5$: CHI + majority rule; $M_6$: CHI + RE algorithm; $M_7$: CHI + H-RE algorithm; $M_8$: iterative imputation; $M_9$: *mice* with cart option.

× types of missing × variables. Thus, we conclude that MICE with CART as assisted models is not suitable for imputing categorical variables.

Second, we focus on all the single-tree methods ($M_1$ to $M_8$) for a fair comparison. Similar to the imputation results for quantitative variables, the imputation accuracy increases when

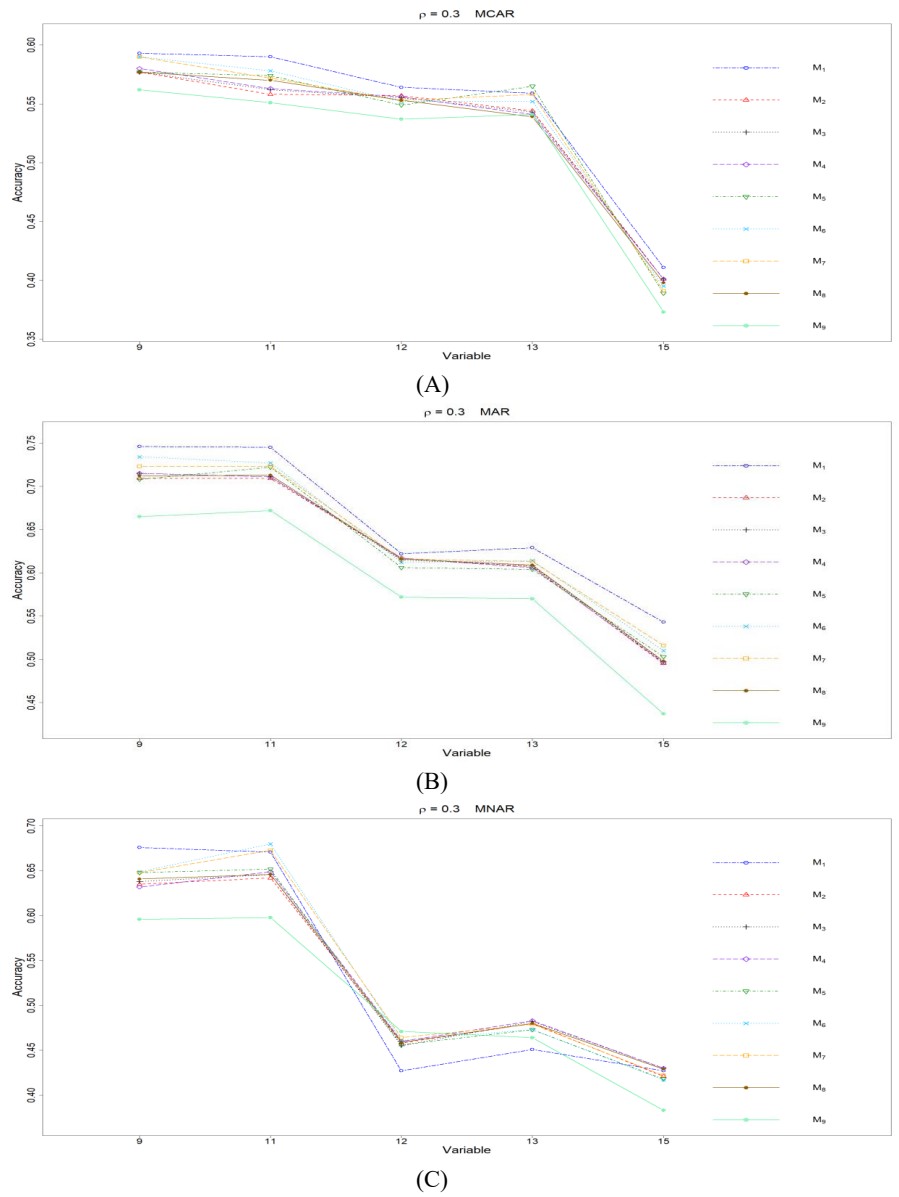

**Figure 6** **The imputation performances for ordinal categorical variables 9, 11, 12, 13, and 15 under the condition correlation = 0.3, missing rate 10%, $N = 500$, and (A) MCAR, (B) MAR, (C) MNAR.** The $PCC_p$ under methods $M_1$ to $M_9$ are plotted respectively for each of the variables. Imputation methods: $M_1$: *rpart* with surrogate variables; $M_2$: GI (RSS) + majority rule; $M_3$: GI (RSS) + RE algorithm; $M_4$: GI (RSS) + H-RE algorithm; $M_5$: CHI + majority rule; $M_6$: CHI + RE algorithm; $M_7$: CHI + H-RE algorithm; $M_8$: iterative imputation; $M_9$: *mice* with cart option.

we increase the correlation $\rho$. In addition, when $\rho > 0$ is combined with MCAR and MAR, $M_1$ generally performs slightly better than do the other methods, as shown in Figs. 6 to 8. Under MNAR, no method outperforms the other methods under various conditions. However, when we concentrate on the best method for each of the conditions, they are generally $M_6$, $M_7$ and $M_1$, according to Figs. 5 to 8. The results highlight the advantages

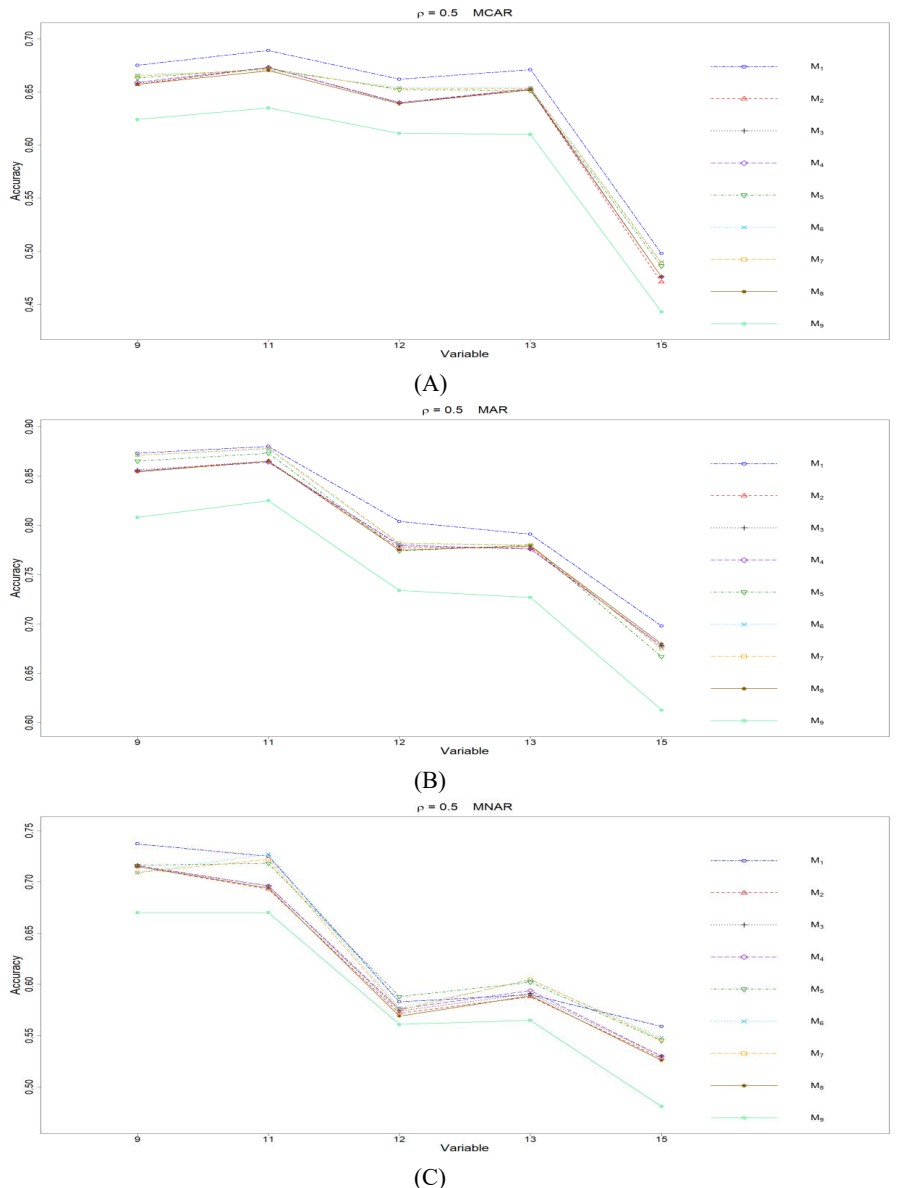

**Figure 7 The imputation performances for ordinal categorical variables 9, 11, 12, 13, and 15 under the condition correlation = 0.5, missing rate 10%, N = 500, and (A) MCAR, (B) MAR, (C) MNAR.** The $PCC_p$ under methods $M_1$ to $M_9$ are plotted respectively for each of the variables. Imputation methods: $M_1$: *rpart* with surrogate variables; $M_2$: GI (RSS) + majority rule; $M_3$: GI (RSS) + RE algorithm; $M_4$: GI (RSS) + H-RE algorithm; $M_5$: CHI + majority rule; $M_6$: CHI + RE algorithm; $M_7$: CHI + H-RE algorithm; $M_8$: iterative imputation; $M_9$: *mice* with cart option.

of the RE or H-RE algorithms. When we fix one of the correlation settings ($\rho = 0.3$, 0.5, or 0.7) and compare $PCC_p$ across the three types of missing mechanisms, we find that the MAR always has a better $PCC_p$ than does its MCAR or MNAR counterpart, which can be interpreted as follows: the missingness of a variable $x_p$ under MAR depends on

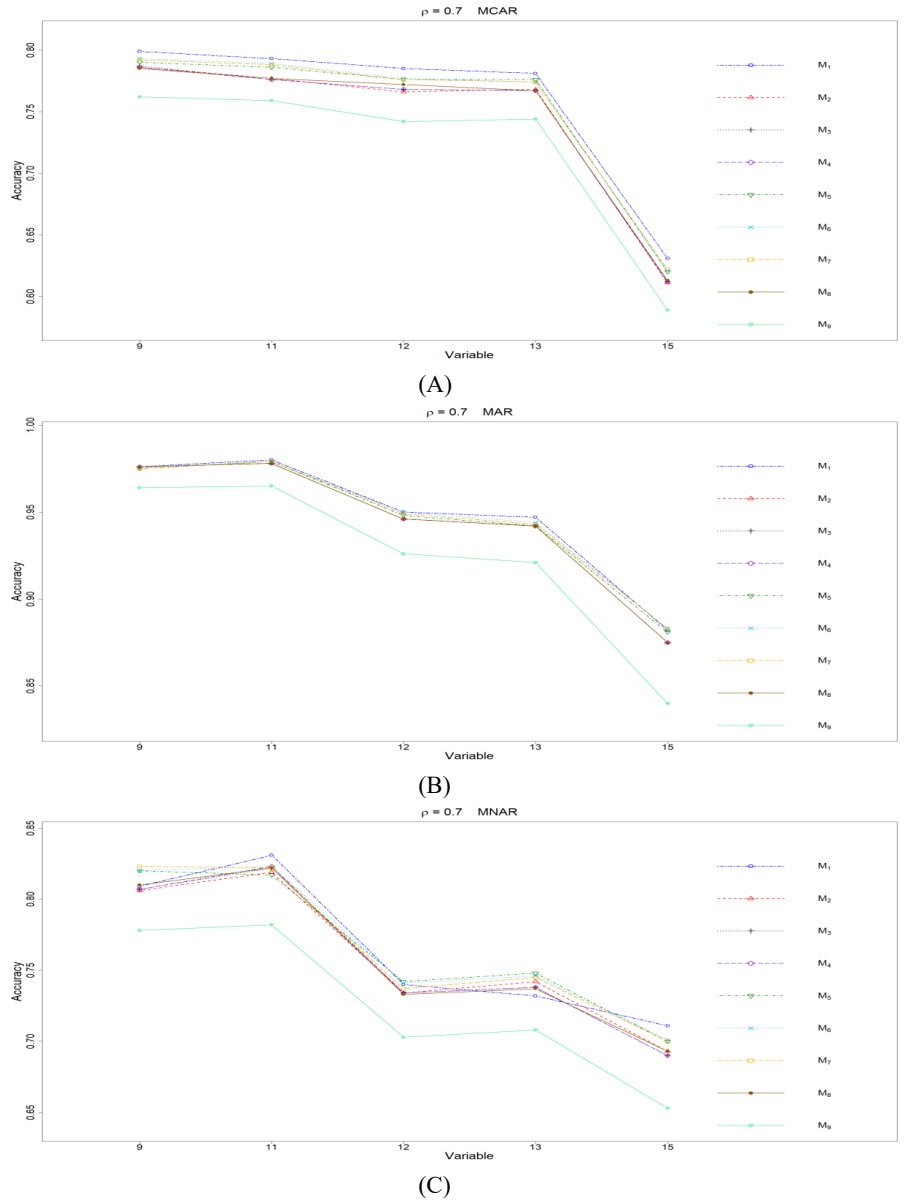

**Figure 8 The imputation performances for ordinal categorical variables 9, 11, 12, 13, and 15 under the condition correlation = 0.7, missing rate 10%, $N = 500$, and (A) MCAR, (B) MAR, (C) MNAR.** The $PCC_p$ under methods $M_1$ to $M_9$ are plotted respectively for each of the variables. Imputation methods: $M_1$: *rpart* with surrogate variables; $M_2$: GI (RSS) + majority rule; $M_3$: GI (RSS) + RE algorithm; $M_4$: GI (RSS) + H-RE algorithm; $M_5$: CHI + majority rule; $M_6$: CHI + RE algorithm; $M_7$: CHI + H-RE algorithm; $M_8$: iterative imputation; $M_9$: *mice* with cart option.

some variable $x_q$ in $\mathcal{S}_p$. In the simulations, we frequently selected $x_q$ as a splitting variable for imputing the missing data on $x_p$; therefore, the imputation accuracy under MAR is generally better.

### Imputation performance for study 2

The imputation performances in terms of $RMSE_p$ under various conditions with the 20% and 30% missing rates are illustrated, respectively, in Figs. S1 to S4 and in S5 to S8. The corresponding mean and standard deviation, over 100 replicates, of $RMSE_p$ are summarized Tables S9 to S16.

We still concentrate on the performance of MICE ($M_9$) *versus* other methods at first. The results under the 20% and 30% missing rates are similar to the results under the 10% missing rate, according to Figs. S1 to S8. As a summary, $M_9$ works better than other methods under $\rho = 0.3$, and it works worse than others under $\rho = 0$.

Second, we focus on the comparison of $M_1$ to $M_8$, as what we did in 'Imputation performance for quantitative variables' and 'Imputation performance for ordinal categorical variables'.

The CHI methods with the RE and the H-RE algorithms ($M_6$ and $M_7$) generally perform well when there is no correlation ($\rho = 0$) and the data are under 20% and 30% missing rates, according to Figs. S1 and S5. The findings are quite consistent with the findings in 'Imputation performance for quantitative variables' under the 10% missing rate. In addition, the GI (RSS) with the H-RE algorithm ($M_4$) also work well under $\rho = 0$ and 20% and 30% missing rates, and it shows the advantage of the H-RE algorithm.

Under the 20% and 30% missing rates and under $\rho = 0.3$ and 0.5, the iterative imputation method ($M_8$) generally performs than the other seven single-tree methods ($M_1$ to $M_7$), according to Figs. S2, S3, S6, and S7. The phenomenon is similar to what we observed under the 10% missing rate. The *rpart* package with surrogate variables ($M_1$) also results in rather good imputations under these correlations.

Quite consistent with our findings under the 10% missing rate, the *rpart* package with surrogate variables ($M_1$), classical GI (RSS) methods ($M_2$ and $M_3$), and the iterative imputation method ($M_8$) all function quite well when $\rho = 0.7$ and missing rates equal 20% and 30%, based on Figs. S4 and S8.

### Imputation performance for study 3

For the condition sample sizes equivalent to 1,000, the mean and standard deviation, taken over 100 replicates, of $RMSE_p$ are given in Table S17 to S20. The settings of Study 3 and those of Study 1 differ in their sample size. The results of Study 3 are generally consistent with the findings under Study 1. We focus on the comparison of the standard deviations in Tables S1 to S4 and S17 to S20. These tables demonstrate nice properties that standard deviations, taken over 100 replicates, of $RMSE_p$ decrease as we increase the sample size for each of the conditions (correlations $\times$ types of missing $\times$ variables). This fits our expectation because of the following reasons. When we increase the sample size, better split variables can be found at the CART-building stage, and therefore, imputations should be more robust over replications. The results suggest us to increase the sample size as possible for imputing one dataset in real applications.

### Real data analysis

We further applied the imputation methods to two real datasets from the Repository of Machine Learning Databases at the University of California Irvine (*Merz & Murphy,*

*1996*): Hepatitis data and Credit approval data. Following *Hapfelmeier, Hothornb & Ulma (2012)*, we considered two studies of missing data imputation to real data. First, for existing missing data, we imputed missing data using the eight methods and compared the imputed values. Second, we further generated additional missing values in the observed datasets, imputed both the existing and generated missing data, and evaluated the imputation performance of the generated missingness. In the real data analysis, we focused on the comparison of single-imputation methods for missing data imputation. Since MICE were some aggregation of multiple imputation values (*Hapfelmeier, Hothornb & Ulma, 2012*), we simply considered $M_1$ to $M_8$ in this section for a fair comparison. The second study differs from the simulation study in 'Simulation studies' in two aspects: (i) in the simulations, we generated the observed data and knew the true underlying distributions and correlations among variables; (ii) in the real data analysis, we encountered two kinds of missing data, and the type of missing mechanisms for the original missingness was unknown.

### Hepatitis data

The Hepatitis data consisted of 155 observations and 20 variables, six of which were quantitative and 14 of which were categorical. The variables included class (the outcome of the disease), age, sex, antivirals, fatigue, liver firm, and so on. There were 15 variables with missing data. For each of variables, we computed the proportions of missing, and the proportions were generally between 0.6% and 7%. There were three variables with higher missing proportions: 10% for albumin, 18% for alk phosphate, and 43% for protime. The correlations between quantitative variables were generally between 0.12 and 0.45 and between −0.12 and −0.45. All 14 categorical variables were binary variables.

We first imputed the original missing data. Interestingly, all eight imputation methods resulted in the same imputation for most of the missing data, and therefore, we omit the results here.

We generated missing data on quantitative variables 15 (bilirubin), 16 (alk phosphate), and 17 (sgot) and generated missing data on categorical variables 4 (steroid), 6 (fatigue), 7 (malaise), 10 (liver firm), and 12 (spiders). The proportions of missing included at this stage was 15%. We repeated the procedure of generating missing data 100 times randomly and independently. Both the original missing data and the generated missing data in each of the 100 resulting datasets were imputed respectively using the eight methods.

For setting the MAR, the missingness of quantitative variables 15 to 17 was arranged to depend on variable 18 since their absolute correlations with variable 18 were higher: the smaller the values of variable 18, the greater the probability of missing for variables 15 to 17. According to the correlations between variables, the missingness on categorical variables 4, 6, 7, 10, and 12 were arranged to depend on variable 9, 8, 8, 9, and 5, respectively. When the dependent variables were in the first category, the probabilities of missing on variables 4, 6, 7, 10, and 12 were set to be greater than those when the dependent variables are in the second category.

For setting MNAR, the probabilities of missing on quantitative variables 15 to 17 were arranged to increase when the original values decreased. For the first category of variables 4, 6, 7, 10, and 12, the probabilities of missing were set to 0.05, 0.25, 0.05, 0.05, and 0.05,

respectively. The probability of missing for the second category were assigned so that generated missing proportion was 15% for each variable.

The imputation accuracy for missingness of quantitative variables is summarized in Table S21. For each of the 100 datasets where we generated missing data under MAR or MNAR, we found out the method that achieved the best imputation among the eight methods and summarized over the 100 datasets in terms of the number of datasets where the method achieved the best performance. The method with the highest number under a condition is marked with a rectangle. There were some ties where more than one method achieved the best results at the same time; therefore, the total number of a row in Table S21 is sometimes greater than 100. The imputation results for the missingness on categorical variables are organized in a similar manner in Table S22.

A method with higher counts is regarded as better. According to the results in Table S21, there is no method that performs better than all the others under all conditions. The CHI methods ($M_5$ to $M_7$) perform better on more datasets for variables 15 and 17. This phenomenon is believed to be related to the fact that the correlations between variables 15–17 are around 0.2. According to the simulation studies, the CHI methods perform well under conditions with lower correlations. The real data are more complicated than the simulated data since we do not know the missing mechanisms of the original missingness; therefore, there might be mixed types of missingness in real data. The iterative method ($M_8$) and the *rpart* package with surrogate variables ($M_1$) also work well on more datasets for variables 15 and 16, respectively. In addition, we do not have to worry about how to send an element with a missing split variable to one of the child nodes in this dataset since none of the three methods is better than the other two methods.

With respect to the imputation to missing categorical variables, there are many ties in the imputation, according to Table S22, and these phenomena are similar to what we found in the simulation studies. Overall, which method works better depends on the variables. The CHI methods ($M_5$ to $M_7$) perform better on variables 4 and 6, while the iterative method ($M_8$) and the *rpart* package with surrogate variables ($M_1$) work better on variables 10 and 12.

To summarize the results for both quantitative and categorical variables, we suggest the following different imputation methods for different variables in the Hepatitis data: the CHI methods for variables 4, 6, and 17; the GI (RSS) methods for variable 7; the iterative method for variable 15; and the *rpart* package with surrogate variables for variables 10, 12, and 16.

### Credit approval data

There are 690 observations and 16 variables in the Credit approval data. Among the 16 variables, six of the variables are quantitative, and the other 10 variables are categorical. Because of confidentiality, the meanings of variables were not released by UCI. We simply used the variable id for our analysis and demonstrated the imputation results. For the six quantitative variables, the absolute values of correlations were generally between 0.2 and 0.4, while some of the correlations were 0.019 and −0.080. Therefore, the correlation structure between variables were more similar to the condition $\rho = 0.3$ in our simulations.

There were seven variables with some missingness, and the proportion of missingness of a variable ranged between 0.9% and 1.9%. Another feature which made the Credit approval data different from the Hepatitis data was that one of the categorical variables (variable 6) was with 14 categories, and there were nine categories in another categorical variable (variable 7).

We generated missing data under the MAR and MNAR mechanisms in the following settings. We generated missing data for quantitative variables on variables 2, 3, and 11; missing data for categorical variables were generated on variables 4, 9, 10, 12, and 13.

To generate MAR, the missingness on variables 2, 3, and 11 were set to be related to the values of variable 8. The probabilities of missing on variables 2, 3, and 11 were set to be higher when the values of variable 8 were smaller. For generating MAR on categorical variables 4, 9, 10, 12, and 13, they were arranged to depend on the values of variables 5, 16, 16, 5, and 5, respectively. When the dependent variables were in the first category, the probabilities of missing on variables 4, 9, 10, 12, and 13 were set to be higher; when the probabilities of missing were set to be smaller when in the second category.

To generate MNAR on quantitative variables 2, 3, and 11, we arranged greater missing probabilities when their original variable values were smaller. For variables 4, 9, 10, 12, and 13, the probabilities of missing on the first category were set to be higher than the probability of missing on the second category. All the remaining settings were the same as the settings for the Hepatitis data in 'Hepatitis data' and were omitted here.

The imputation results for the generated missingness on quantitative variables and categorical variables are reported in Tables S23 to S24, respectively, in terms of the number of datasets, among the 100 generated missing datasets, where the method achieved the best performance. The *rpart* package with surrogate variables ($M_1$) performs better on some variables (variables 2 and 11), based on Table S23. The results consist with what we observed for variable 16 in Hepatitis data. Regarding the imputation results for categorical variables, there are many ties, and the imputation performances still depend on variables. For variables 4 and 10, most methods perform equally well. The iterative method ($M_8$) performs better on variable 9. In addition, some imputation performances not only depend on variables but also depend on the types of missing mechanisms: $M_1$ performs better under MAR on variable 12 and under MNAR on variable 13; CHI methods ($M_6$ and $M_7$) work well under MAR on variable 13; $M_2$ performs slightly better under MNAR on variable 12.

To sum up the imputation studies of the two real data, the imputation of real data are rather complicated. None of the eight methods can outperform all the others, and the imputation sometimes not only depend on variables but also depend on the types of missing mechanisms. Since we do not know the types of missing mechanisms for the original missing data in the real data, there might be mixture types of missing mechanisms in our imputation studies for real data. Combined with what we learned in the simulation studies, the imputation results for real data are some combined effects of different correlations between variables, different types of missing mechanisms, and the ratios of categories for one categorical variable.

# DISCUSSION AND CONCLUSIONS

In the present study, we compared several CART-based missing data imputation methods under various conditions and aimed to find out the method(s) with best imputation accuracy. The methods in comparison included the *rpart* package with surrogate variables, GI (RSS) methods, CHI methods, the proposed resampling algorithms, iterative imputation methods, and MICE. Overall, we present the following findings, contributions, and suggestions. To our limited knowledge, this is the first study to bring all these methods together and to examine some conditions for comparing imputation efficiency. An important finding is that the method that performs the best strongly depends on the correlation between variables, which is rarely considered and examined in the literature. For example, *Hapfelmeier, Hothornb & Ulma (2012)* and *Rahman & Islam (2013)* conducted the typical missing data studies where real data were adopted in numerical studies so that correlations between variables could not be manipulated as a factor in their studies. *Ramosaj & Pauly (2019)* concentrated on a specific correlation in their simulation settings. *Xu, Daniels & Winterstein (2016)* focused on various regression-type relationship between variables in their simulation, and the correlations between variables were in a more complex structure. The implication of our findings: future researches about the CART-based missing data imputation are recommended to include the correlation between variables as one factor in simulation studies. To future real data analysis, we would recommend to choose imputation methods depending on the correlation between variables so that better imputation accuracy can be achieved.

In addition, we also propose a new perspective on missing data in a CART imputation problem and realize the perspective in the RE and H-RE algorithms in the current study. We find that the CHI methods with RE and H-RE algorithms are useful when correlations between quantitative variables are close to 0. The results provide future data analysis some useful message: when the correlations between variables are low, CHI methods with RE and H-RE algorithms are suggested for imputation because of their imputation accuracy.

For imputing missing ordinal categorical variables, MICE ($M_9$) generally performs worse than the other eight methods and thus does not be recommended. When we concentrate on single-tree methods for a fair comparison, the *rpart* package with surrogate variables ($M_1$) is recommended under $\rho > 0$ with MCAR and MAR conditions, based on our simulation results. Similar findings have been shown in *Hapfelmeier, Hothornb & Ulma (2012)*, where they examined the imputation efficiency on real data (with unknown correlations and missing mechanisms) and found that the *rpart* package functioned well, compared to other single-tree imputation methods. Under MNAR, $M_6$, $M_7$ and $M_1$ are suggested.

For imputing missing quantitative variables, we start with the condition under 10% missing rate and summarize findings in the followings. These findings provide useful guidelines to future imputation applications. The imputation methods one chooses should depend on the correlations between variables, in order to achieve higher imputation accuracy. MICE ($M_9$) is recommended under correlations around 0.3 ($\rho = 0.3$). The iterative imputation method ($M_8$) is most recommended under $\rho = 0.5$ since it is robust across various variables and missing mechanisms. There were some related findings

in literature, where *Stekhoven & Bühlmann (2012)* also found that iterative imputation methods generally performed well in various datasets. We would like to emphasize one key difference between their settings and those in the current study: they concentrated on real data so that the correlations between could not be controlled in their numerical study. In the current study, we find that the imputation performance of one method actually depends on the correlations between variables, and the iterative imputation method generally work well under moderate correlation conditions. In addition, *Hapfelmeier, Hothornb & Ulma (2012)* found that MICE was especially suitable in MAR settings. Our findings suggest that the imputation performance of MICE depends on the correlations between variables. The linkage to literatures further highlights the need to consider the correlations between variables in further researches. The imputation accuracy of the commonly used method, the *rpart* package with surrogate variables ($M_1$), is slightly worse than that of $M_8$ under moderate correlation conditions but is also acceptable. When the correlation is increased to a high level ($\rho = 0.7$), the RMSE of $M_1$, $M_8$, classical GI (RSS) methods ($M_2$ to $M_4$), and $M_9$ are at some comparable scales, and all of them are acceptable. When there is no correlation ($\rho = 0$), the CHI methods $M_5$ to $M_7$ generally perform better than the other six methods. *Loh & Shih (1997)* and *Kim & Loh (2001)* remarked that their proposed CHI method is selection unbiased when there is no correlation among variables and when there are no missing data in the dataset. In the present study, we further found out that the CHI methods $M_5$ to $M_7$ also outperform the other six methods when missing data exist. How to send elements with missing split variables to one of the child nodes is an issue. According to the results in Fig. 1 to Fig. 4, when we fix one method for selecting split variables (GI, RSS, or CHI), the resampling methods (RE or H-RE algorithm; $M_3$, $M_4$, $M_6$, $M_7$) generally work better under $\rho = 0$, while the majority methods ($M_2$, $M_5$) generally function better under $\rho = 0.3$ and $0.7$. We suggest selecting among the majority rule, the RE algorithm, or the H-RE algorithm according to the correlations between variables. What's more, all the above phenomena are consistent across different missing rates and sample sizes, according to simulation studies 2 and 3.

On the basis that the imputation efficiency depends on the correlation between variables, we suggest first computing correlations between variables and then choosing one imputation method with the best imputation accuracy based on the resulting variable correlations when analyzing real data. For ordinal categorical variables, the type of missing mechanism is further required to select one good imputation method. Given that the type of missing data mechanism may be unknown in real data, we would suggest selecting a more robust method. In specific, under $\rho > 0$, we can use $M_1$ for categorical variables and $M_8$ and $M_9$ for quantitative variables because its imputation accuracy under various conditions is generally in good or moderate levels. Under $\rho$ close to 0, CHI methods are recommended.

In the present study, we followed the framework that treats each variable with missingness as a response variable and sequentially impute it (*Rahman & Islam, 2013*; *Stekhoven & Bühlmann, 2012*). Other CART-based imputation methods exist, for example, the proximity imputation in *Breiman (2003)* and *Tang & Ishwaran (2017)*. They considered the similarity between subjects in a so-called proximity matrix and imputed missing

data using a subject with the highest similarity or using a weighted average of observed values. Their work presents another interesting framework for imputation, and all of the investigations in the current study can be further applied in the framework.

## ACKNOWLEDGEMENTS

We thank the editor and reviewers for valuable comments and suggestions which improve this article. We are grateful to Dr. Yu-Chung Wei for his valuable suggestions.

### Funding
This work was supported by the National Science and Technology Council (grant number MOST 110-2118-M-004-005 and NSTC 112-2118-M-004-003). The funders had no role in study design, data collection and analysis, decision to publish, or preparation of the manuscript.

### Grant Disclosures
The following grant information was disclosed by the authors:
National Science and Technology Council:  MOST 110-2118-M-004-005,  NSTC 112-2118-M-004-003.

### Competing Interests
The authors declare there are no competing interests.

### Author Contributions
- Cheng-Yang Chen performed the experiments, analyzed the data, performed the computation work, prepared figures and/or tables, and approved the final draft.
- Yu-Wei Chang conceived and designed the experiments, analyzed the data, authored or reviewed drafts of the article, and approved the final draft.

### Data Availability
The code are available in the Supplementary Files.

The Hepatitis data are available at the Repository of Machine Learning Databases at the University of California Irvine: Hepatitis. (1988). UCI Machine Learning Repository. https://doi.org/10.24432/C5Q59J.

The Credit approval data are available at the Repository of Machine Learning Databases at the University of California Irvine: Quinlan,J. R.. Credit Approval. UCI Machine Learning Repository. https://doi.org/10.24432/C5FS30.

### Supplemental Information
Supplemental information for this article can be found online at http://dx.doi.org/10.7717/peerj-cs.2119#supplemental-information.

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
