# Peer review of "Missing data imputation using classification and regression trees"

_PeerJ Computer Science, doi:10.7717/peerj-cs.2119_

## Round 0.1 · original submission · Major Revisions

Please, address the concerns raised by the reviewers, specially those related to the experimentation included in the paper.

**Language Note:** The review process has identified that the English language must be improved. PeerJ can provide language editing services - please contact us at [email protected] for pricing (be sure to provide your manuscript number and title). Alternatively, you should make your own arrangements to improve the language quality and provide details in your response letter. – PeerJ Staff

Reviewer 1 ·

Basic reporting

Summary: A new perspective on CART-based imputation is explored, involving resampling algorithms to enhance the imputation process. The paper compares various CART-based methods to identify the most effective approaches under different conditions through simulation studies and apply the methods to real datasets for illustration.

Comments:
- State the research question the manuscript is trying to answer and briefly describe its primary contributions (e.g., proposing two resampling-based methods for missing data imputation using CART). This could be done in the paragraph starting on line 92 in the Introduction.
- Briefly describe the two resampling based methods in the Introduction; e.g., “Algorithm 1 (RE Algorithm) performs random assignments for elements with missing values on the splitting variable to either child node multiple times and selects the best split based on impurity measures to improve CART model fitting. Algorithm 2 (H-RE Algorithm) extends the first by conducting separate resampling for each node and its children, ensuring that the resampling of a parent node does not directly determine the resampling for its child nodes, thus maintaining a hierarchy in the decision-making process.”
- Briefly summarize the key findings after line 100 in the Introduction.
Number the figures in the order in which they are referenced in the main text.
- In Tables 1-4, which report the results of various imputation methods (M1 through M8), include a brief description of each imputation method in the table, either in the column header or table caption.
Similarly, in Figures 1-8 include a brief description of each imputation method (M1 through M8), in the legend or caption.

Experimental design

The experimental design for the simulation studies in the paper is robust and detailed. It addresses the performance of imputation methods under varying conditions of data, which includes different correlations and missing data mechanisms. The inclusion of both quantitative and ordinal categorical variables provides a comprehensive assessment across common data types. The reproducibility of the simulations is strong due to the clear description of the simulation settings, the structured approach to generating data, and the use of well-defined missing mechanisms (MCAR, MAR, MNAR).

Validity of the findings

The findings of the study are valid and contribute significantly to the field of missing data imputation with ML methods. By comparing CART-based imputation methods under various conditions, the study highlights the dependency of imputation efficiency on the correlation between variables. The study's recommendation of specific imputation methods based on correlation levels and missing data mechanisms provides practical guidelines for future applications. Moreover, the introduction of RE and H-RE algorithms offers a new perspective on addressing missing data in CART imputation, enhancing the effectiveness of CHI methods for variables with low correlations.

Additional comments

The paper would benefit from a more detailed explanation of why certain methods are recommended for specific conditions. Specifically, when discussing the recommendation of CHI methods for variables with correlations close to zero (line 486), it could be useful to include a theoretical or empirical rationale that supports this choice. Discuss how the structure of CHI methods and the selection criteria used by RE and H-RE algorithms specifically advantage low correlation scenarios.

In Section 4, discuss situations where the RE or H-RE algorithms might not perform as expected, for example, in datasets where variables are highly correlated. In such cases, the resampling strategy might not capture the complex relationships between variables as effectively as other methods designed for high correlation environments.

Cite this review as

Reviewer 2 ·

Basic reporting

The descriptions of the two proposed algorithms (RE and H-RE) are impossible to follow. Instead of using lay language, it is better to present them in pseudo code. Besides, it will be helpful to indicate what ``RE'' stands for.

Experimental design

Imputation of missing data is already implemented in the mice R package. Its cart option ought to be included for comparison.

The simulation experiments are too simple. All quantitative variables being multivariate normal and all ordinal categorical variables seem to be independent.

The missingness mechanisms in the simulations are too simple as well. The MNAR setting should be more realistic.

Validity of the findings

The findings are limited by the rather simple simulation setups. The simulation results in the figures need to show simulation standard errors.

Cite this review as

Reviewer 3 ·

Basic reporting

The essay adheres strongly to professional standards, using clear and straightforward language throughout and offering enough prior literature to frame the study. The format is well-organized, with useful graphics and tables that successfully supplement the text. Raw data access is consistent with journal policies, improving reproducibility. While the results are cohesive and explicitly address the research objectives, the authors may increase the impact of the publication by emphasizing the uniqueness and significance of their findings, especially in the discussion and conclusion sections. Offering insights into practical consequences and suggestions for future research would enhance the study's overall contribution to the field of missing data imputation.

Experimental design

The research described in the submission fits the standards of original primary research within the scope of the journal, with a well-defined and significant research issue that answers a known knowledge gap. To improve the presentation, make more explicit linkages between the study question, methods, and findings throughout the paper. Furthermore, improving the clarity and organization of the methods section may aid in better comprehension and replication of the work. Including visual aids like figures or tables to illustrate significant findings or data trends could also improve the presentation and make the results more understandable to readers. Furthermore, maintaining uniformity in vocabulary and layout throughout the book would result in a more polished and professional presentation. Overall, by applying these recommendations,the manuscript could achieve greater clarity and effectiveness in communicating its research findings.

Validity of the findings

The manuscript recognizes the significance of analyzing impact and novelty, emphasizing the usefulness of significant replication studies that add to the research. Encourage replication with clearly articulated rationales and advantages, which matches with the journal's rigorous research criteria. However, to develop the work further, the authors should explicitly address the possible significance and novelty of their findings in relation to the existing literature. Giving readers a clear reason for the study's significance and how it increases knowledge in the subject might help them grasp its importance. Furthermore, ensuring that all underlying data are robust, statistically sound, and stored or available in an appropriate repository is critical for transparency and repeatability.

Additional comments

It would be useful to investigate the applicability of the suggested methodologies in various datasets or contexts for subsequent research. This can entail putting the techniques to the test on a range of datasets with various attributes, including sample sizes, variable kinds, and missingness levels. Furthermore, examining the approaches' resilience to various missing data processes, apart from those examined in this work, would offer significant understanding into their applicability. Moreover, carrying out comparative analyses with alternative imputation techniques—including those excluded from the present analysis—may facilitate a more thorough identification of the advantages and disadvantages of each strategy. Lastly, more sophisticated and customized imputation procedures may result from taking into account the possible influence of variables or outside factors on missingness patterns and imputation accuracy.

Cite this review as

---

## Round 0.2 · Minor Revisions

The reviewer has stated that adding a couple of references could benefit the paper. However, the policy of PeerJ is that authors should only add them if they actually think that they are relevant and could help to improve the context of the work.

Please, take a look to the references and, if you consider that they (or one of them) are relevant, add them to the paper.

Reviewer 1 ·

Basic reporting

no comment

Experimental design

no comment

Validity of the findings

no comment

Additional comments

This reviewer is satisfied with the authors' responses and revisions made to the manuscript.

Since MICE with CART as the imputer is now included in the comparison, it would benefit the paper to cite the following two studies in Section 2.1.2. The first shows that MICE with CART outperforms deep generative models for missing data imputation in survey data; the second shows MICE with CART comprehensively outperforms MICE with GLM.

- Wang, Zhenhua, et al. "Are deep learning models superior for missing data imputation in surveys? Evidence from an empirical comparison." Survey Methodology 48 (2022): 375-399.

- Akande, Olanrewaju, Fan Li, and Jerome Reiter. "An empirical comparison of multiple imputation methods for categorical data." The American Statistician 71.2 (2017): 162-170.

Cite this review as

---

## Round 0.3 · accepted · Accept

Authors have addressed all the comments that were made by the reviewers, the editor and the staff of the journal.